# ZERO-SHOT LEARNING
# WITH COMMON SENSE KNOWLEDGE GRAPHS

## ABSTRACT

Zero-shot learning relies on semantic class representations such as hand-engineered attributes or learned embeddings to predict classes without any labeled examples. We propose to learn class representations from common sense knowledge graphs. Common sense knowledge graphs are an untapped source of explicit high-level knowledge that requires little human effort to apply to a range of tasks. To capture the knowledge in the graph, we introduce ZSL-KG, a general-purpose framework with a novel transformer graph convolutional network (TrGCN) to generate class representations. Our proposed TrGCN architecture computes non-linear combinations of the node neighbourhood and leads to significant improvements on zero-shot learning tasks. We report new state-of-the-art accuracies on six zero-shot benchmark datasets in object classification, intent classification, and fine-grained entity typing tasks. ZSL-KG outperforms the specialized state-of-the-art method for each task by an average 1.7 accuracy points and outperforms the general-purpose method with the best average accuracy by 5.3 points. Our ablation study on ZSL-KG with alternate graph neural networks shows that our transformer-based aggregator adds up to 2.8 accuracy points improvement on these tasks.

## 1 INTRODUCTION

Deep neural networks require large amounts of labeled training data to achieve optimal performance. This is a severe bottleneck, as obtaining large amounts of hand-labeled data is an expensive process. Zero-shot learning is a training strategy which allows a machine learning model to predict novel classes without the need for any labeled examples for the new classes (Romera-Paredes & Torr, 2015; Socher et al., 2013; Wang et al., 2019). Zero-shot models learn parameters for seen classes along with their class representations. During inference, new class representations are provided for the unseen classes. Previous zero-shot learning systems have used hand-engineered attributes (Akata et al., 2015; Farhadi et al., 2009; Lampert et al., 2014), pretrained embeddings (Frome et al., 2013) and learned embeddings (e.g. sentence embeddings) (Xian et al., 2016) as class representations.

Class representations in a zero-shot learning framework should satisfy the following properties: (1) they should adapt to unseen classes without requiring additional human effort, (2) they should provide rich features such that the unseen classes have sufficient distinguishing characteristics among themselves, (3) they should be applicable to a range of downstream tasks. Previous approaches for class representations have various limitations. On one end of the spectrum, attribute-based methods provide rich features but the attributes have to be fixed ahead of time for the unseen classes. On the other end of the spectrum, pretrained embeddings such as GloVe (Pennington et al., 2014) and Word2Vec (Mikolov et al., 2013) offer the flexibility of easily adapting to new classes but rely on unsupervised training on large corpora—which may not provide distinguishing characteristics necessary for zero-shot learning. Many methods lie within the spectrum and learn class representations for zero-shot tasks from descriptions such as attributes, text, and image prototypes. Existing approaches that have achieved state-of-the-art performance make task-specific adjustments and cannot exactly be adapted to tasks in different domains (Liu et al., 2019a; Verma et al., 2020). Methods using graph neural networks on the ImageNet graph to learn class representations have achieved strong performance on zero-shot object classification (Kampffmeyer et al., 2019; Wang et al., 2018). These methods are general-purpose, since we show that they can be adapted to other tasks as well. However, the ImageNet graph may not provide rich features suitable for a wide range of downstream tasks.

In our work, we propose to learn class representations from common sense knowledge graphs. Common sense knowledge graphs (Liu & Singh, 2004; Speer et al., 2017; Tandon et al., 2017; Zhang et al., 2020) are an untapped source of explicit high-level knowledge that requires little human effort to apply to a range of tasks. These graphs have explicit edges between related concept nodes and provide valuable information to distinguish between different concepts. However, adapting existing zero-shot learning frameworks to learn class representations from common sense knowledge graphs is challenging in several ways. GCNZ (Wang et al., 2018) learns graph neural networks with a symmetrically normalized graph Laplacian, which not only requires the entire graph structure during training but also needs retraining if the graph structure changes, i.e., GCNZ is not inductive. Common sense knowledge graphs can be large (2 million to 21 million edges) and training a graph neural network on the entire graph can be prohibitively expensive. DGP (Kampffmeyer et al., 2019) is an inductive method and aims to generate expressive class representations but assumes a directed acyclic graph such as WordNet. Common sense knowledge graphs do not have a directed acyclic graph structure.

To address these limitations, we propose ZSL-KG, a general-purpose framework with a novel transformer graph convolutional network (TrGCN) to learn class representations. Graph neural networks learn to represent the structure of graphs by aggregating information from each node's neighbourhood. Aggregation techniques used in GCNZ, DGP, and most other graph neural network approaches are linear, in the sense that they take a (possibly weighted) mean or maximum of the neighbourhood features. To capture the complex information in the common sense knowledge graph, TrGCN learns a transformer-based aggregator to compute the non-linear combination of the node neighbours. A few prior works have considered LSTM-based aggregators (Hamilton et al., 2017a) as a way to increase the expressivity of graph neural networks, but their outputs can be sensitive to the ordering imposed on the nodes in each neighborhood. For example, on the Animals with Attributes 2 dataset, we find that when given the same test image 10 times with different neighborhood orderings, an LSTM-based graph neural network outputs inconsistent predictions 16% of the time (Appendix A). One recent work considers trying to make LSTMs less sensitive by averaging the outputs over permutations, but this significantly increases the computational cost and provides only a small boost to prediction accuracy (Murphy et al., 2019). In contrast, TrGCN learns a transformer-based aggregator, which is non-linear and naturally permutation invariant. Additionally, our framework is inductive, i.e., the graph neural network can be executed on graphs that are different from the training graph, which is necessary for inductive zero-shot learning under which the test classes are unknown during training.

We demonstrate the effectiveness of our framework on three zero-shot learning tasks in vision and language: object classification, intent classification, and fine-grained entity typing. We report new state-of-the-art accuracies on six zero-shot benchmark datasets (Xian et al., 2018a; Farhadi et al., 2009; Deng et al., 2009; Coucke et al., 2018; Gillick et al., 2014; Weischedel & Brunstein, 2005). ZSL-KG outperforms the state-of-the-art specialized method for each task by an average 1.7 accuracy points. ZSL-KG also outperforms GCNZ, the best general-purpose method on average by 5.3 accuracy points. Our ablation study on ZSL-KG with alternate graph neural networks shows that our transformer-based aggregator adds up to 2.8 accuracy points improvement on these tasks.

In summary, our main contributions are the following:

1. We propose to learn class representations from common sense knowledge graphs for zero-shot learning.
2. We present ZSL-KG, a general-purpose framework based on graph neural networks with a novel transformer-based architecture. Our proposed architecture learns non-linear combination of the nodes neighbourhood and generates expressive class representations.
3. ZSL-KG achieves new state-of-the-art accuracies on Animals with Attributes 2 (Xian et al., 2018a), aPY (Farhadi et al., 2009), ImageNet (Deng et al., 2009), SNIPS-NLU (Coucke et al., 2018), Ontonotes (Gillick et al., 2014), and BBN (Weischedel & Brunstein, 2005).

## 2 BACKGROUND

In this section, we summarize zero-shot learning and graph neural networks.

## 2.1 ZERO-SHOT LEARNING

Zero-shot learning has several variations (Wang et al., 2019; Xian et al., 2018a). Our work focuses on inductive zero-shot learning, under which we do not have access to the unseen classes during training. We train a zero-shot classifier by optimizing over the seen classes. But, unlike traditional methods, zero-shot classifiers are trained along with class representations such as attributes, pretrained embeddings, etc.

Recent approaches learn a class encoder $\phi(y) \in \mathbb{R}^d$ to produce vector-valued class representations from an initial input, such as a string or other identifier of the class. (In our case, $y$ is a node in a graph and its $k$-hop neighborhood.) During inference, the class representations are used to label examples with the unseen classes by passing the examples through an example encoder $\theta(x) \in \mathbb{R}^d$ and predicting the class whose representation has the highest inner product with the example representation.

Recent work in zero-shot learning commonly uses one of two approaches to learn the class encoder $\phi(y)$. One approach uses a bilinear similarity function defined by a compatibility matrix $\boldsymbol{W} \in \mathbb{R}^{d \times d}$ (Frome et al., 2013; Xian et al., 2018a):

$$f\left(\theta(x), \boldsymbol{W}, \phi(y)\right) = \theta(x)^{\mathrm{T}} \boldsymbol{W} \phi(y) . \tag{1}$$

The bilinear similarity function gives a score for each example-class pair. The parameters of $\theta$, $\boldsymbol{W}$, and $\phi$ are learned by taking a softmax over $f$ for all possible seen classes $y \in Y_S$ and minimizing either the cross entropy loss or a ranking loss with respect to the true labels. In other words, $f$ should give a higher score for the correct class(es) and lower scores for the incorrect classes. $\boldsymbol{W}$ is often constrained to be low rank, to reduce the number of learnable parameters (Obeidat et al., 2019; Yogatama et al., 2015). Lastly, other variants of the similarity function add minor variations such as non-linearities between factors of $\boldsymbol{W}$ (Socher et al., 2013; Xian et al., 2016).

The other common approach is to first train a neural network classifier in a supervised fashion. The final fully connected layer of this network has a vector representation for each seen class, and the remaining layers are used as the example encoder $\theta(x)$. Then, the class encoder $\phi(y)$ is trained by minimizing the L2 loss between the representations from supervised learning and $\phi(y)$ (Kampffmeyer et al., 2019; Socher et al., 2013; Wang et al., 2018).

The class encoder that we propose in Section 3 can be plugged into either approach.

## 2.2 GRAPH NEURAL NETWORKS

The basic idea behind graph neural networks is to learn node embeddings that reflect the structure of the graph (Hamilton et al., 2017b). Consider the graph $\mathcal{G} = (V, E, R)$, where $V$ is the set of vertices with node features $X_v$ and $(v_i, r, v_j) \in E$ are the labeled edges and $r \in R$ are the relation types. Graph neural networks learn node embeddings by iterative aggregation of the k-hop neighbourhood. Each layer of a graph neural network has two main components AGGREGATE and COMBINE (Xu et al., 2019):

$$\boldsymbol{a}_v^{(l)} = \text{AGGREGATE}^{(l)} \left( \left\{ \boldsymbol{h}_u^{(l-1)} \, \forall u \in \mathcal{N}(v) \right\} \right) \tag{2}$$

where $\boldsymbol{a}_v^{(l)} \in \mathbb{R}^{d_{l-1}}$ is the aggregated node feature of the neighbourhood, $\boldsymbol{h}_u^{(l-1)}$ is the node feature in neighbourhood $\mathcal{N}(.)$ of node $v$ including a self loop. The aggregated node is passed to the COMBINE to generate the node representation $\boldsymbol{h}_v^{(l)} \in \mathbb{R}^{d_l}$ for the $l$-th layer:

$$\boldsymbol{h}_v^{(l)} = \text{COMBINE}^{(l)} \left( \boldsymbol{h}_v^{(l-1)}, \boldsymbol{a}_v^{(l)} \right) \tag{3}$$

$\boldsymbol{h}_v^{(0)} = \boldsymbol{x}_v$ where $\boldsymbol{x}_v$ is the initial feature vector for the node. Previous works on graph neural networks for zero-shot learning have used GloVe (Pennington et al., 2014) to represent the initial features (Kampffmeyer et al., 2019; Wang et al., 2018).

## 3 THE ZSL-KG FRAMEWORK

Here we introduce ZSL-KG: a general-purpose framework with a novel Transformer Graph Convolutional Network to learn class representation from commmon sense knowledge graphs. Figure 1 shows the ZSL-KG architecture with an example `elephant` concept.

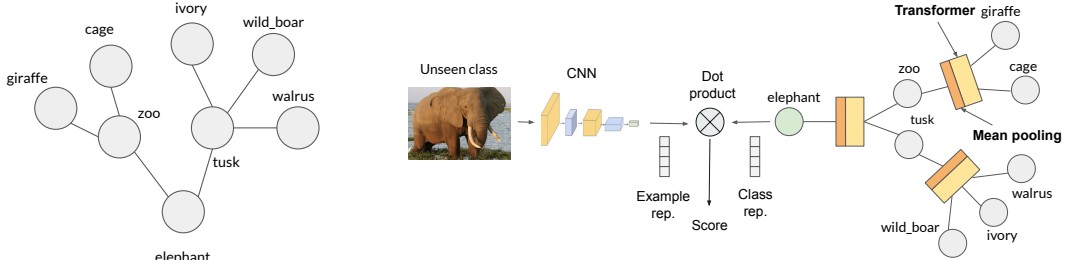

Figure 1: *Left:* A sample from the 2-hop neighbourhood for the concept `elephant` from ConceptNet. *Right:* The figure describes the architecture for ZSL-KG. The image of an elephant is passed through the example encoder (ResNet101) to generate the example representation for the image. For the class representation, we take the sampled k-hop neighbourhood and pass the nodes through their hop-specific Transformer Graph Convolutional Network. We take the dot product of the example representation and the class representation to compute the score for the class. The same architecture is followed for language tasks with a task-specific example encoder.

### 3.1 COMMON SENSE KNOWLEDGE GRAPHS FOR ZERO-SHOT LEARNING.

Common sense knowledge graphs organize high-level knowledge implicit to humans in a graph. The nodes in the graph are concepts associated with each other via edges. These associations in the graph offer a rich source of information, which makes them applicable to a wide range of tasks. Publicly available common sense knowledge graphs range roughly from 100,000 to 8 million nodes and 2 million to 21 million edges (Speer et al., 2017; Zhang et al., 2020). To learn class representations from common sense knowledge graphs, we look to graph neural networks.

Existing zero-shot learning methods such as GCNZ (Wang et al., 2018) and DGP (Kampffmeyer et al., 2019) that learn class representations from structured knowledge are applicable only to small graphs such as ImageNet or Wordnet as they make restrictive assumptions. GCNZ requires the entire graph structure during training, and learns class representations from the ImageNet graph which is significantly smaller than common sense knowledge graphs. For instance, the ImageNet graph used in GCNZ has about 32,000 nodes and 65,000 edges. DGP learns a more expressive class representation but requires a directed acyclic graph or parent-child relationship in the graph. We are not restricted to parent-child relationships in a common sense knowledge graph.

### 3.2 TRANSFORMER GRAPH CONVOLUTIONAL NETWORK.

To overcome these limitations, we propose to learn class representations with a novel graph neural network: transformer graph convolutional networks (TrGCN). Transformers (Vaswani et al., 2017) are non-linear modules typically used for machine translation and language modeling tasks. They achieve a non-linear combination of the input sequences using two-layer feedforward neural networks and scaled dot product attention. We exploit this property to learn a non-linear aggregator that captures the complex structure of a common sense knowledge graph. Finally, they are not tied to the graph structure which makes them well suited for inductive zero-shot learning.

Here we describe TrGCN. We pass the neighbhourhood node features $h_u^{(l-1)}$ through a two-layer feedforward neural network with a ReLU activation between the layers. The previous features are added to its output features with a skip connection, followed by layer normalization (Ba et al., 2016):

$$h_u'^{(l-1)} = \text{LayerNorm}\left\{ \boldsymbol{W}_{fh}^{(l)} \left[ \text{ReLU}\left( \boldsymbol{W}_{hf}^{(l)} \boldsymbol{h}_u^{(l-1)} \right) \right] + \boldsymbol{h}_u^{(l-1)} \ \forall u \in \mathcal{N}(v) \right\} \quad (4)$$

where $\boldsymbol{W}_{hf}^{(l)} \in \mathbb{R}^{d_{(l-1)} \times d_{(f)}}$ and $\boldsymbol{W}_{fh}^{(l)} \in \mathbb{R}^{d_{(f)} \times d_{(l-1)}}$ are learnable weight matrices for the feedforward neural network. The non-linear neighbourhood features are then passed through the scaled dot product attention layer to compute the weighted combination of the features for each query node:

$$\left\{ \boldsymbol{z}_u^{(l)} \ \forall u \in \mathcal{N}(v) \right\} = \text{softmax}\left( \frac{\boldsymbol{Q}\boldsymbol{K}^T}{\sqrt{d_{(p)}}} \right) \boldsymbol{V} \quad (5)$$

where $\boldsymbol{Q} = \boldsymbol{W}_q^{(l)} \boldsymbol{h}_u'^{(l-1)}$ is the set of all neighbourhood query vectors, $\boldsymbol{K} = \boldsymbol{W}_k^{(l)} \boldsymbol{h}_u'^{(l-1)}$ is the set of all key vectors, $\boldsymbol{V} = \boldsymbol{W}_v^{(l)} \cdot \boldsymbol{h}_u'^{(l-1)}$ is the set of values vectors, and $\boldsymbol{W}_q \in \mathbb{R}^{d_{(l-1)} \times d_{(p)}}$, $\boldsymbol{W}_k \in \mathbb{R}^{d_{(l-1)} \times d_{(p)}}$, $\boldsymbol{W}_v \in \mathbb{R}^{d_{(l-1)} \times d_{(p)}}$ are learnable weight matrices with the projection dimension $d_{(p)}$. The output features from the attention layer is projected with another linear layer and added to its previous features with a skip connection, followed by layer normalization:

$$\left\{ \boldsymbol{z}_u'^{(l)} \; \forall u \in \mathcal{N}(v) \right\} = \text{LayerNorm} \left( \boldsymbol{W}_z^{(l)} \boldsymbol{z}_u^{(l-1)} + \boldsymbol{h}_u'^{(l-1)} \; \forall u \in \mathcal{N}(v) \right) \tag{6}$$

where $\boldsymbol{W}_z^{(l)} \in \mathbb{R}^{d_{(p)} \times d_{(l-1)}}$ is a learnable weight matrix.

To get the aggregated vector $\boldsymbol{a}_v^{(l)}$ for node $v$, we pass the output vectors $\{ \boldsymbol{z}_u'^{(l)} \; \forall u \in \mathcal{N}(v) \}$ from the transformer through a permutation invariant pooling function $\mu(.)$ such as mean-pooling. The aggregated vector is passed through a linear layer followed by a non-linearity $\sigma(.)$ such as ReLU or LeakyReLU:

$$\boldsymbol{a}_v^{(l)} = \mu \left( \left\{ \boldsymbol{z}_u'^{(l)} \; \forall u \in \mathcal{N}(v) \right\} \right) \quad \boldsymbol{h}_v^{(l)} = \sigma \left( \boldsymbol{W}^{(l)} \cdot \boldsymbol{a}_v^{(l)} \right) \tag{7}$$

where $\boldsymbol{W}^{(l)} \in \mathbb{R}^{d_{(l-1)} \times d_{(l)}}$ is a learnable weight matrix.

Existing work has drawn parallels between transformers and graph attention networks (GAT), suggesting that they are equivalent (Joshi, 2020). GAT (Veličković et al., 2018) computes the aggregated vector by taking the linear combination of the node features in the neighbourhood. In contrast, TrGCN learns a transformer-based aggregator and computes a non-linear combination of the node features in the neighbourhood, followed by a pooling function to get the aggregated vector which leads to the difference in architecture.

**Neighbourhood Sampling.** In our experiments, we use ConceptNet (Speer et al., 2017) as our common sense knowledge graph but our approach applies to other knowledge graphs. ConceptNet has high node degree, which poses a challenge to train the graph neural network. To solve this problem, we explored numerous neighbourhood sampling strategies. Existing work on sampling neighbourhood includes random sampling (Hamilton et al., 2017a), importance sampling (Chen et al., 2018a), random walks (Ying et al., 2018), etc. Similar to PinSage (Ying et al., 2018), we simulate random walks for the nodes in the graph and assign hitting probabilities to the neighbourhood nodes. During training and testing the graph neural network, we select the top $N$ nodes from the neighbourhood based on their hitting probability.

## 4 TASKS AND RESULTS

We evaluate our framework on three zero-shot learning tasks: object classification, intent classification, and fine-grained entity typing. In all our experiments, we compare ZSL-KG with the state-of-the-art specialized methods for the task as well as general-purpose methods: GCNZ, SGCN, and DGP. The code and hyperparameters are included in the supplementary material, which will be released upon acceptance.

**Setup.** We use two-layer graph neural networks for the general-purpose baselines and ZSl-KG. We use the code from Kampffmeyer et al. (2019) and adapt GCNZ (Wang et al., 2018), SGCN, and DGP (Kampffmeyer et al., 2019) for zero-shot learning tasks in language (See Appendix C.1). In all our experiments with ZSL-KG, we map each class to a node in ConceptNet 5.7 (Speer et al., 2017) and query its 2-hop neighbourhood. We simulate random walks for each node in the ConceptNet graphs and compute the hitting probability for the nodes in the neighbourhood. Then, we sample 50 and 100 node neighbours with the highest hitting probabilities for the first and the second hop, respectively. See Appendix C.2 for more details.

### 4.1 OBJECT CLASSIFICATION

Object classification is a computer vision task of categorizing objects into different class categories.

**Datasets.** We evaluate on the Animals with Attributes 2 (AWA2) (Xian et al., 2018a), attribute Pascal Yahoo (aPY) (Farhadi et al., 2009), and ImageNet datasets (Deng et al., 2009). AWA2 contains images of animals with 40 classes in the train set and 10 classes in the test set. aPY contains images

|  | *AWA2* | *aPY* |
|---|---|---|
|  | Accuracy | Accuracy |
| SP-AEN | 59.7 | 37.0 |
| LisGAN | 55.8 | 36.8 |
| ZSML | 77.5 | 64.0 |
| GCNZ | $77.00 \pm 2.05$ | $56.02 \pm 0.63$ |
| SGCN | $77.76 \pm 1.60$ | $54.98 \pm 0.66$ |
| DGP | $76.26 \pm 1.69$ | $52.22 \pm 0.49$ |
| ZSL-KG | $\mathbf{78.08 \pm 0.84}$ | $\mathbf{64.36 \pm 0.59}$ |

Table 1: Results for object classification on the AWA2 and aPY dataset. We report the average class-balanced accuracy of the models on 5 random seeds and the standard error. The results for SP-AEN, LisGAN, and ZSML are obtained from Verma et al. (2020).

|  | *SNIPS-NLU* |
|---|---|
|  | Accuracy |
| Zero-shot DNN | 71.16 |
| IntentCapsNet | 77.52 |
| ReCapsNet-ZS | 79.96 |
| GCNZ | $82.47 \pm 03.09$ |
| SGCN | $50.27 \pm 14.13$ |
| DGP | $64.41 \pm 12.87$ |
| ZSL-KG | $\mathbf{88.98 \pm 01.22}$ |

Table 2: Results for intent classification on the SNIPS-NLU dataset. We report the average accuracy of the models on 5 random seeds and the standard error. The results for Zero-shot DNN, IntentCapsNet, and ReCapsNet-ZS are obtained from Liu et al. (2019a).

| | | Hit@k(%) | | | | | | | | | | | | | |
|---|---|---|---|---|---|---|---|---|---|---|---|---|---|---|---|
| | | 2-Hops | | | | | 3-Hops | | | | | All | | | | |
| | | 1 | 2 | 5 | 10 | 20 | 1 | 2 | 5 | 10 | 20 | 1 | 2 | 5 | 10 | 20 |
| ZSL | GCNZ | 19.8 | 33.3 | 53.2 | 65.4 | 74.6 | 4.1 | 7.5 | 14.2 | 20.2 | 27.7 | 1.8 | 3.3 | 6.3 | 9.1 | 12.7 |
| | SGCN | 26.2 | 40.4 | 60.2 | 71.9 | 81.0 | 6.0 | 10.4 | 18.9 | 27.2 | 36.9 | 2.8 | 4.9 | 9.1 | 13.5 | 19.3 |
| | DGP | 26.6 | 40.7 | 60.3 | 72.3 | 81.3 | 6.3 | 10.7 | 19.3 | 27.7 | 37.7 | 3.0 | 5.0 | 9.3 | 13.9 | 19.8 |
| | ZSL-KG | **26.8** | **41.3** | **61.5** | **73.2** | **82.0** | **6.6** | **11.4** | **20.8** | **30.0** | **39.7** | **3.1** | **5.5** | **10.3** | **15.3** | **21.7** |
| GZSL | GCNZ | 9.7 | 20.4 | 42.6 | 57.0 | 68.2 | 2.2 | 5.1 | 11.9 | 18.0 | 25.6 | 1.0 | 2.3 | 5.3 | 8.1 | 11.7 |
| | SGCN | **11.9** | **27.0** | 50.8 | 65.1 | 75.9 | 3.2 | 7.1 | 16.1 | 24.6 | 34.6 | 1.5 | 3.4 | 7.8 | 12.3 | 18.2 |
| | DGP | 10.3 | 26.4 | 50.3 | 65.2 | 76.0 | 2.9 | 7.1 | 16.1 | 24.9 | 35.1 | 1.4 | 3.4 | 7.9 | 12.6 | 18.7 |
| | ZSL-KG | 11.5 | 26.5 | **51.2** | **65.8** | **76.5** | **3.5** | **7.7** | **17.5** | **27.0** | **37.4** | **1.8** | **3.9** | **8.8** | **13.9** | **20.5** |

Table 3: Results for object classification on ImageNet dataset. We report the class-balanced top-k accuracy on zero-shot learning (ZSL) and generalized zero-shot learning (GZSL) for ImageNet classes k-hops away from the ILSVRC 2012 classes. The results for GCNZ, SGCN, and DGP are obtained from Kampffmeyer et al. (2019).

of objects with 20 classes in the train set and 12 classes in the test set. ImageNet contains images from 1000 classes in the train set and 21K classes in the test set.

**Experiment.** Following prior work (Kampffmeyer et al., 2019; Wang et al., 2018), here we use the L2 loss architecture for zero-shot learning. The example encoder and seen class representations come from the ResNet 101 model (He et al., 2016) in Torchvision (Marcel & Rodriguez, 2010) pretrained on ILSVRC 2012 (Russakovsky et al., 2015). We map the ILSVRC 2012 training and validation classes, and the AWA2 and aPY test classes to ConceptNet. The model is trained for 1000 epochs on 950 random classes and the remaining 50 ILSVRC 2012 classes are used for validation. We use the same setting for GCNZ, SGCN, and DGP using the authors' implementation. The model with the least loss on the validation classes is used to make predictions on the test classes. For the ImageNet experiment, we train the model for 3000 epochs on 1000 classes from ILSVRC 2012 with the common sense knowledge graph and switch to ImageNet graph during test time. Similar to DGP, we freeze the final layer with the generated class representations and fine-tune the ResNet-backbone on the ILSVRC images for 15 epochs using SGD with a learning rate 0.0001 and momentum of 0.9. Following prior work, we report the class-balanced accuracy (Xian et al., 2018a) on unseen classes for AWA2 and aPY. We follow the train/test split from (Frome et al., 2013), and evaluate ZSL-KG on two levels of difficulty: 2-hops, and 3-hops. The hops refer to the distance of the classes from the ILSVRC train classes. We evaluate ZSL-KG on two settings: zero-shot learning (ZSL) where only unseen classes are present and generalized zero-shot learning (GZSL) where both seen and unseen classes are present. Following previous work (Kampffmeyer et al., 2019) on ImageNet evaluation, we report the class-balanced top-K accuracy.

We compare ZSL-KG against state-of-the-art specialized methods for AWA2 and aPY: SP-AEN (Chen et al., 2018b), LisGAN (Li et al., 2019), and ZSML (Verma et al., 2020). ZSML is a GAN-based approach and has reported the highest results on AWA2 and aPY, and DGP and SGCN have reported the highest on results on ImageNet.

**Results.** Table 1 shows the results for zero-shot object classification. ZSL-KG outperforms existing state-of-the-art methods on the AWA2 and aPY datasets. The general-purpose methods show a significant drop in accuracy from AWA2 to aPY, whereas our method consistently achieves the highest accuracy on both the datasets. This suggests that the class representations generated from a richer graph help in performance. In contrast to specialized methods, ZSL-KG trains only one model and does not require any specialized training datasets to achieve state-of-the-art performance on both the datasets. Finally, our experiments with ImageNet show that ZSL-KG, despite being trained on a noisier graph, reports the state-of-the-art on zero-shot learning and generalized zero-shot learning. ZSL-KG offers accuracy improvements up to 2.3 points and relative improvements up to 20% over the state-of-the-art methods.

## 4.2 INTENT CLASSIFICATION

To assess ZSL-KG's versatility, we experiment on zero-shot intent classification. Intent classification is a text classification task of identifying users' intent expressed in chatbots and personal voice assistants.

**Dataset.** We evaluate on the main open-source benchmark for intent classification: SNIPS-NLU (Coucke et al., 2018). The dataset was collected using crowdsourcing to benchmark the performance of voice assistants. The training set has 5 seen classes which we split into 3 train classes and 2 development classes.

**Experiment.** Zero-shot intent classification is a multi-class classification task. The example encoder used in our experiments is a biLSTM with attention (Appendix F). We train the model for 10 epochs by minimizing the cross entropy loss and pick the model with the least loss on the development set. We measure accuracy on the test classes.

We compare ZSL-KG against existing specialized state-of-the-art methods in the literature for zero-shot intent classification: Zero-shot DNN (Kumar et al., 2017), IntentCapsNet (Xia et al., 2018), and ResCapsNet-ZS (Liu et al., 2019a). IntentCapsNet and ResCapsNet-ZS are CapsuleNet-based (Sabour et al., 2017) approaches and have reported the best performance on the task.

**Results.** Table 2 shows the results. ZSL-KG significantly outperforms the existing approaches and improves the state-of-the-art accuracy to 88.98%. The general-purpose methods have mixed performance on intent classification and suggests ZSL-KG works well on a broader range of tasks.

## 4.3 FINE-GRAINED ENTITY TYPING

To test ZSL-KG's ability to classify fine-grained types, we experiment on zero-shot fine-grained entity typing. Fine-grained entity typing is the task of classifying named entities into one or more narrowly scoped semantic types. This task also tests generalized zero-shot learning when seen classes appear in the test set.

**Datasets.** We evaluate on popular fine-grained entity typing datasets: OntoNotes (Gillick et al., 2014) and BBN (Weischedel & Brunstein, 2005).

We split the dataset into two: coarse-grained labels and fine-grained labels. Following the prior work (Obeidat et al., 2019), we train on the coarse-grained labels and predict on both coarse-grained and fine-grained labels in the test set. Details about the datasets can be found in Appendix B.

**Experiment.** Fine-grained entity typing is a zero-shot multi-label classification task because each entity can be associated with more than one type. We reconstructed OTyper (Yuan & Downey, 2018) and DZET (Obeidat et al., 2019), the state-of-the-art specialized methods for this task. Both methods use the AttentiveNER biLSTM (Shimaoka et al., 2017) as the example encoder. See Appendix G for more details.

We train each model for 5 epochs by minimizing the cross-entropy loss. During inference, we pass the scores from the bilinear similarity model through a sigmoid and pick the labels that have probability

|  | *Ontonotes* | | | *BBN* | | |
|---|---|---|---|---|---|---|
|  | Strict Acc. | Loose Mic. | Loose Mac. | Strict Acc. | Loose Mic. | Loose Mac. |
| OTyper | $41.72 \pm 0.44$ | $55.00 \pm 0.37$ | $61.25 \pm 0.53$ | $25.76 \pm 0.25$ | $54.75 \pm 0.68$ | $62.88 \pm 0.67$ |
| DZET | $42.88 \pm 0.47$ | $56.16 \pm 0.41$ | $62.26 \pm 0.46$ | $26.20 \pm 0.13$ | $\mathbf{57.31 \pm 0.24}$ | $\mathbf{63.44 \pm 0.30}$ |
| GCNZ | $41.46 \pm 0.81$ | $54.61 \pm 0.52$ | $61.65 \pm 0.52$ | $21.47 \pm 1.22$ | $47.17 \pm 1.04$ | $56.81 \pm 0.68$ |
| SGCN | $42.64 \pm 0.69$ | $\mathbf{56.25 \pm 0.29}$ | $62.93 \pm 0.39$ | $24.91 \pm 0.24$ | $50.02 \pm 0.73$ | $59.68 \pm 0.54$ |
| DGP | $41.11 \pm 0.78$ | $54.08 \pm 0.62$ | $61.38 \pm 0.76$ | $23.99 \pm 0.14$ | $47.19 \pm 0.51$ | $57.62 \pm 0.89$ |
| ZSL-KG | $\mathbf{45.21 \pm 0.36}$ | $55.81 \pm 0.41$ | $\mathbf{63.64 \pm 0.51}$ | $\mathbf{26.69 \pm 2.41}$ | $50.30 \pm 1.21$ | $57.66 \pm 1.15$ |

Table 4: The results for zero-shot fine-grained entity typing on Ontonotes and BBN. We report the average strict accuracy, loose micro F1, and loose macro F1 of the models on 5 random seeds and the standard error.

|  | *AWA2* | *aPY* | *SNIP-NLU* | *OntoNotes* | *BBN* |
|---|---|---|---|---|---|
|  | Accuracy | Accuracy | Accuracy | Strict Acc. | Strict Acc. |
| ZSL-KG-GCN | $74.81 \pm 0.93$ | $61.63 \pm 0.54$ | $84.78 \pm 0.77$ | $42.19 \pm 0.75$ | $27.44 \pm 2.94$ |
| ZSL-KG-GAT | $75.29 \pm 0.64$ | $62.87 \pm 0.36$ | $87.57 \pm 1.59$ | $43.48 \pm 0.91$ | $\mathbf{32.65 \pm 2.27}$ |
| ZSL-KG-RGCN | $66.27 \pm 1.01$ | $54.20 \pm 0.63$ | $87.47 \pm 1.81$ | $43.88 \pm 0.85$ | $27.89 \pm 1.47$ |
| ZSL-KG-LSTM | $66.46 \pm 1.19$ | $54.89 \pm 0.74$ | $88.81 \pm 1.17$ | $44.52 \pm 0.22$ | $26.77 \pm 0.63$ |
| ZSL-KG | $\mathbf{78.08 \pm 0.84}$ | $\mathbf{64.36 \pm 0.59}$ | $\mathbf{88.98 \pm 1.22}$ | $\mathbf{45.21 \pm 0.36}$ | $26.69 \pm 2.41$ |

Table 5: The results for zero-shot learning tasks with other graph neural networks. We report the average performance on 5 random seeds and the standard error.

of 0.5 or greater as our prediction. As is common for this task (Obeidat et al., 2019), we evaluate the performance of our model on strict accuracy, loose micro F1, and loose macro F1 (Appendix H). Strict accuracy penalizes the model for incorrect label predictions and the number of the label predictions have to match the ground truth, whereas loose micro F1 and loose macro F1 measures if the correct label is predicted among other false positive predictions.

**Results.** Table 4 shows the results. ZSL-KG outperforms the existing state-of-the-art specialized methods on strict accuracy for both OntoNotes and BBN. DZET has higher loose micro on both the datasets because it overpredicts labels and has greater false positives compared to the ZSL-KG. ZSL-KG has higher precision for label predictions and therefore, higher strict accuracy compared to other methods. These results demonstrate that our method works well even in a generalized multilabel setting where the test set has multiple seen and unseen labels.

## 4.4 Comparison of Graph Aggregators

We conduct an ablation study with different aggregators with our framework. Existing graph neural networks include GCN (Kipf & Welling, 2017), GAT (Veličković et al., 2018), RGCN (Schlichtkrull et al., 2018) and LSTM (Hamilton et al., 2017a). We provide all the architectural details in Appendix I. We train these models with the same experimental setting for the tasks mentioned in their respective sections.

**Results.** Table 5 shows results for our ablation study. Our results show that ZSL-KG almost always outperforms existing graph neural networks with linear aggregators. ZSL-KG-LSTM which uses an LSTM-based aggregator shows inconsistent performance across different tasks indicating that they are more useful on low-dimensional tasks such as node classification. With relational aggregators (ZSL-KG-RGCN), we observe that they do not outperform ZSL-KG and may reduce the overall performance (as seen in AWA2 and aPY). Finally, it is worth comparing SGCN and ZSL-KG-GCN as they use the same linear aggregator to learn the class representation but train on different graphs. We see that ZSL-KG-GCN trained on common sense knowledge graphs adds an average improvement of 8.1 accuracy points across the tasks suggesting that the choice of graphs is crucial for downstream performance.

## 5 RELATED WORK

We broadly describe related works on zero-shot learning and graph neural networks.

**Zero-Shot Learning.** Zero-shot learning has been thoroughly researched in the computer vision community for object classification (Akata et al., 2015; Farhadi et al., 2009; Frome et al., 2013; Lampert et al., 2014; Wang et al., 2019; Xian et al., 2018a). Recent works in zero-shot learning have used graph neural networks for object classification (Kampffmeyer et al., 2019; Wang et al., 2018). In our work, we extend their approach to common sense knowledge graphs to generate class representations. Furthermore, we learn TrGCN, a novel graph neural network with a non-linear aggreagtor to learn the structure of common sense knowledge graphs. Recent work on zero-shot object classification has used attention over image regions to achieve strong results on the task (Zhu et al., 2019; Liu et al., 2019b; Xie et al., 2019; Liu et al., 2020). In contrast, our work focuses on the class encoder (TrGCN), where we learn attention over the graph and could potentially complement methods that focus on image encoders. Recently work on zero-shot learning has focused on applying attention over the image regions. (Zhu et al., 2019; Liu et al., 2019b; Xie et al., 2019; Liu et al., 2020) but they focus on the example encoder, particularly, they apply attention over the image regions. Other notable works use generative methods for generalized zero-shot learning where both seen and unseen classes are evaluated at test time (Kumar Verma et al., 2018; Schonfeld et al., 2019). However, these methods still rely on hand-crafted attributes for classification. Zero-shot learning has been studied in text classification as well (Dauphin et al., 2013; Nam et al., 2016; Pappas & Henderson, 2019; Zhang et al., 2019b; Yin et al., 2019). Previously, ConceptNet has been used for transductive zero-shot text classification as shallow features for class representation (Zhang et al., 2019b). They use ConceptNet to generate a sparse vector which is combined with pretrained embeddings and natural language description to obtain the class representation. On the other hand, we use ConceptNet to generate dense vector representations from a graph neural network and use them as our class representation. Finally, zero-shot fine-grained entity typing has been previously studied with a variety of class representations (Ma et al., 2016; Obeidat et al., 2019; Yuan & Downey, 2018). We note that ZOE (Zhou et al., 2018) is a specialized method for zero-shot fine-grained entity typing and achieves best results on the task. However, they use a subset of the test set that contains both seen and unseen types to tune the threshold parameters which reveals information about the unseen types and makes ZOE a transductive zero-shot learning method.

**Graph Neural Networks.** Recent work on graph neural networks has demonstrated significant improvements for several downstream tasks such as node classification and graph classification (Hamilton et al., 2017a;b; Kipf & Welling, 2017; Veličković et al., 2018; Wu et al., 2019). Extensions of graph neural networks to relational graphs have produced significant results in several graph-related tasks (Marcheggiani & Titov, 2017; Schlichtkrull et al., 2018; Shang et al., 2019; Vashishth et al., 2020). Previous work has used transformers with graph neural networks as a method to generate meta-paths in the graph rather than as a neighbourhood aggregation technique (Yun et al., 2019). A related work Zhang et al. (2019a) combines common sense knowledge graph and graph neural networks for zero-shot learning. ZSL-KG learns to map nodes in a knowledge graph to class representations without any other human input. In contrast, Zhang et al. (2019a) learns to transfer hand-engineered attributes to new nodes in the graph. On the other hand, our work learns a graph neural network over the common sense knowledge graph to generate rich class representations. Finally, several diverse applications using graph neural networks have been explored: common sense reasoning (Lin et al., 2019), fine-grained entity typing (Xiong et al., 2019), text classification (Yao et al., 2019), reinforcement learning (Adhikari et al., 2020) and neural machine translation (Bastings et al., 2017). For a more in-depth review, we point readers to Wu et al. (2020).

## 6 CONCLUSION

ZSL-KG is a flexible framework for zero-shot learning with common sense knowledge graphs and can be adapted to a wide range of tasks without requiring additional human effort. Our framework introduces a novel transformer graph convolutional network (TrGCN) which captures the complex associations in the graph to learn class representations. We achieve state-of-the-art performance on five benchmark datasets across three zero-shot tasks. Our work demonstrates that common sense knowledge graphs are a powerful source of high-level knowledge and can benefit a range of tasks.

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

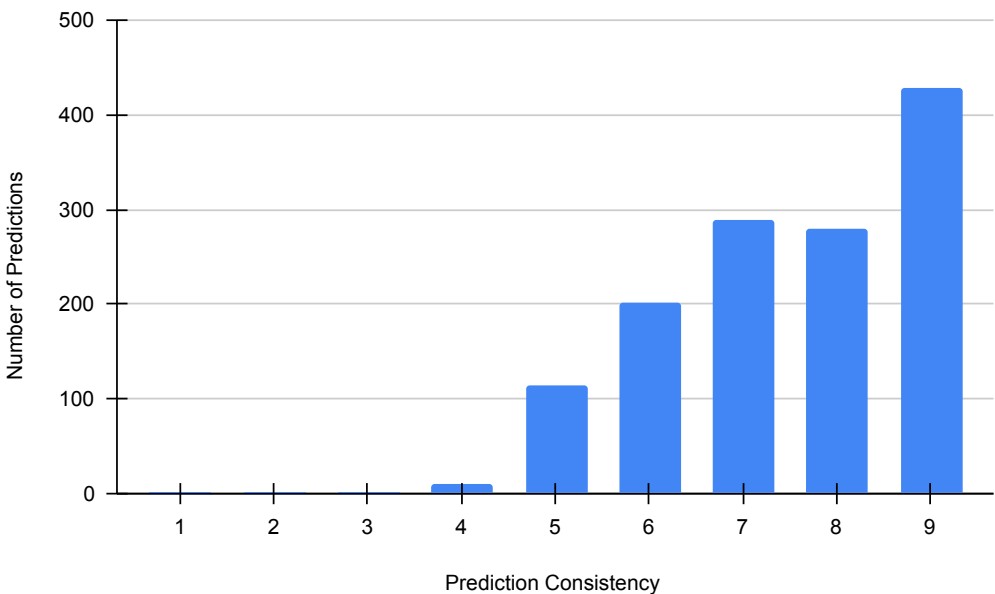

Figure 2: Graph showing distribution of inconsistencies in the LSTM-based aggreagtor predictions.

## A  LSTM Predictions

LSTMs have been used in graph neural networks as aggregators to generate more expressive node embeddings. However, LSTMs assume an ordering of the inputs which is not present in a graph neighbourhood. To apply LSTMs to an unordered set of nodes, Hamilton et al. (2017a) randomly permute the nodes in the neighbourhood.

We test whether randomly permuting the node neighbours makes LSTM-based aggregator permutation invariant. We replicate the LSTM-based aggregator to work with the ZSL-KG framework. The model is trained with the setup described in Section 4.1. We run the prediction on the Animals with Attributes 2 dataset by computing 10 class representation for each of the classes using the trained LSTM-based aggregator model.

The experiments reveal that 1325 out of 7913 (16.78%) have multiple predictions for the same image. For images that have multiple predictions, we take the count of the mode prediction and plot the histogram. Figure 2 shows inconsistency in predictions. The graph for a given value $p$ on the x-axis is read as for every 10 prediction, $p$ times the same output is predicted.

## B  Dataset Details

Here, we provide details about the datasets used in our experiments. Table 6 shows the statistics about the datasets.

We obtain the OntoNotes and BBN dataset from Ren et al. (2016). OntoNotes has three levels of types such as /location, /location/structure, /location/structure/government where /location and /location/structure are treated as coarse-grained entity types and /location/structure/government is treated as fine-grained entity type. Similarly, BBN has two levels of types and we consider the level two types as fine-grained types. Furthermore, we process the datasets by removing all the fine-grained labels from the train set. We also remove all the from the train set where coarse-grained entity types are not present in the test set. Another key point to remember is that /other type present in the OntoNotes dataset cannot be mapped to a meaningful concept or a wikipedia article. To overcome this limitation, we learn an embedding for /other type during training.

| Dataset | Seen classes | Unseen classes | Train examples | Test examples |
|---|---|---|---|---|
| AWA2 | 40 | 10 | - | 7913 |
| aPY | 20 | 12 | - | 7924 |
| SNIPS-NLU | 5 | 2 | 9888 | 3914 |
| OntoNotes | 40 | 32 | 220398 | 9604 |
| BBN | 15 | 24 | 85545 | 12349 |

Table 6: Zero-shot datasets used in our experiments

For object classification datasets, AWA2 and aPY, we do not require the training examples because we use pretrained weights from ResNet101 to learn the class representations. We crop objects from APY test dataset as multiple objects are present in the same image. To crop the objects, we use bounding box provided in Farhadi et al. (2009), add 15px padding on all sides and crop them.

## C   EXPERIMENTAL SETUP

### C.1   GRAPH NEURAL NETWORK SETUP

GCNZ (Wang et al., 2018) uses symmetrically normalized graph Laplacian to generate the class representations. SGCN (Kampffmeyer et al., 2019) uses an asymmetrical normalized graph Laplacian to learn the class representations. Finally, DGP (Kampffmeyer et al., 2019) exploits the hierarchical graph structure and avoids dilution of knowledge from intermediate nodes. They use a dense graph connectivity scheme with a two-stage propagation from ancestors and descendants to learn the class representations.

### C.2   CONCEPTNET SETUP

We further preprocess the queried graph from ConceptNet for all the datasets. We remove all the non-English concepts and their edges from the graph and make all the edges bidirectional. For fine-grained entity typing and object classification, we also take the union of the concepts' neighbourhood that share the same prefix. For example, we take the union of the `/c/en/elephant` and `/c/en/elephant/n`. Then, we compute the embeddings for the concept using the pretrained GloVe 840B (Pennington et al., 2014). We average the individual word in the concept to get the embedding. These embeddings serve as initial features for the graph neural network.

For the random walk, the number of steps is 20, and the number of restarts is 10. We add one smoothing to the visit counts and normalize the counts for the neighboring nodes.

### C.3   FINE-GRAINED ENTITY TYPING SETUP

We reconstructed OTyper (Yuan & Downey, 2018) and DZET (Obeidat et al., 2019) for fine-grained entity typing. Otyper averages the GloVe embeddings for the words in the name of each class to represent it. For DZET, we manually mapped the classes to Wikipedia articles. We pass each article's first paragraph through a learnable biLSTM with attention to obtain the class representations (Appendix F).

## D   PSEUDOCODE

Our framework can be trained in two ways - (1) joint training (2) class encoder training. Joint training means that our example encoder and class encoder is trained jointly with the task. We use joint training for intent classification and fine-grained entity typing. Class encoder training means that we use a pretrained example encoder such as ResNet101 and train only the class encoder. In object classification, we train only the class encoder and use a pretrained ResNet101. In Algorithm 1, we describe the forward pass with the ZSL-KG framework.

---

**Algorithm 1:** Forward pass with the ZSL-KG framework

---

**Input** : example $x$, example encoder $\theta(x)$, linear layer $\boldsymbol{W}$, class encoder $\phi(y)$, graph
$\mathcal{G}(V, E, R)$, class nodes $\{v_y^1, v_y^2, ..., v_y^n\}$, node initialization $\boldsymbol{H} = [h^0, ..., h^v]$, depth $L$,
graph neural network weights $\{\boldsymbol{W}^1, ..., \boldsymbol{W}^L\}$, transformers $\{T^1, ..., T^L\}$,
neighbourhood sample sizes $\{s_1, , ..., s_L\}$,

**Output :** logits for classes $y = \{y_1, y_2, ..., y_n\}$

TrGCN($\mathbb{V}$, $l = 0$) ;

**if** $l = L$ **then**
| return $\boldsymbol{H}_\mathbb{V} \leftarrow \boldsymbol{H}(\mathbb{V})$;

**else**
| $i \leftarrow 0$;
| **for** $v \in \mathbb{V}$ **do**
| | $\mathbb{N} \leftarrow \mathcal{N}(v, s_{l+1})$;
| | $\boldsymbol{H}_\mathbb{N} \leftarrow$ TrGCN($\mathbb{N}$, $l + 1$) ;
| | $\boldsymbol{Z}_\mathbb{N} \leftarrow T^{(l+1)}(\boldsymbol{H}_\mathbb{N})$;
| | $\boldsymbol{a}_v^{(l+1)} \leftarrow \text{mean}(\boldsymbol{Z}_\mathbb{N})$;
| | $\boldsymbol{h}_v^{(l+1)} \leftarrow \sigma(\boldsymbol{W}^{(l+1)} \cdot \boldsymbol{a}_v)$;
| | $\boldsymbol{H}_i^{(l+1)} = \boldsymbol{h}_v^{(l+1)}/||\boldsymbol{h}_v^{(l+1)}||_2$;
| | $i \leftarrow i + 1$
| **end**
| return $\boldsymbol{H}^{(l+1)}$

**end**

$\phi(y) \leftarrow$ TrGCN($\{v_y^1, v_y^2, ..., v_y^n\}$) ;

return $\theta(x)^{\text{T}} \boldsymbol{W} \phi(y)$;

---

## E  OBJECT CLASSIFICATION RESULTS ON AWA2 AND aPY

Table 7 shows results comparing ZSL-KG with other related work from zero-shot object classification.

## F  BiLSTM WITH ATTENTION

The biLSTM with attention is used as the example encoder in intent classification and as class representation for DZET (Obeidat et al., 2019). BiLSTM with attention has two components: (1) biLSTM model (2) attention model. The input tokens $\boldsymbol{w} = \boldsymbol{w}_0, \boldsymbol{w}_1, ..., \boldsymbol{w}_n$ represented by GloVe are passed through the biLSTM to get the hidden states $\overrightarrow{\boldsymbol{h}}$ and $\overleftarrow{\boldsymbol{h}}$. The hidden states are concatenated to get $\boldsymbol{h} = \boldsymbol{h}_0, \boldsymbol{h}_1, ..., \boldsymbol{h}_n$. They are passed through the attention module to get the scalar attention values and normalize them:

$$\alpha_i = \boldsymbol{W}_\alpha \left( \tanh(\boldsymbol{W}_e \cdot \boldsymbol{h}_i) \right) \tag{8}$$

$$a_i = \frac{\exp\left(\alpha_i\right)}{\sum_i \exp\left(\alpha_i\right)} \tag{9}$$

The scalar values are multiplied with their respective hidden vectors to get the final representation $\boldsymbol{t}_w$:

$$\boldsymbol{t}_w = \sum_{i=0}^{n} a_i \boldsymbol{h}_i \tag{10}$$

## G  ATTENTIVENER

We describe AttentiveNER (Shimaoka et al., 2017) that we use to represent the example in the fine-grained entity typing task. Each mention $m$ comprises of $n$ tokens mapped to a pretrained word embedding from GloVe. We average the embeddings to obtain a single vector $\boldsymbol{v}_m$:

$$\boldsymbol{v}_m = \frac{1}{n} \sum_{j=1}^{n} \boldsymbol{m}_j \tag{11}$$

|  | AWA2 | aPY |
|---|---|---|
|  | Accuracy | Accuracy |
| LATEM*(Xian et al., 2016) | 55.8 | 35.2 |
| ESZSL* (Romera-Paredes & Torr, 2015) | 58.6 | 38.3 |
| SYNC* (Changpinyo et al., 2016) | 46.6 | 23.9 |
| DEM* (Zhang et al., 2017) | 67.1 | 35.0 |
| ZSKL* (Zhang & Koniusz, 2018) | 70.5 | 45.3 |
| SP-AEN* (Chen et al., 2018b) | 59.7 | 37.0 |
| CVAE-ZSL* (Mishra et al., 2018) | 71.4 | - |
| f-CLSWGAN* (Xian et al., 2018b) | 68.2 | - |
| LisGAN* (Li et al., 2019) | 55.8 | 36.8 |
| ZSML* (Verma et al., 2020) | 77.5 | 64.0 |
| AREN (Xie et al., 2019) | 67.9 | 39.2 |
| LGGAA (Liu et al., 2019b) | 68.0 | 41.3 |
| SGMAL (Zhu et al., 2019) | 68.8 | - |
| TGG (Zhang et al., 2019a) | 77.2 | 63.5 |
| APNet (Liu et al., 2020) | 68.0 | - |
| E-PGN (Yu et al., 2020) | 73.4 | - |
| APN (Xu et al., 2020) | 73.8 | - |
| GCNZ Wang et al. (2018) | $77.00 \pm 2.05$ | $56.02 \pm 0.63$ |
| SGCN (Kampffmeyer et al., 2019) | $77.76 \pm 1.60$ | $54.98 \pm 0.66$ |
| DGP (Kampffmeyer et al., 2019) | $76.26 \pm 1.69$ | $52.22 \pm 0.49$ |
| ZSL-KG (Ours) | $\mathbf{78.08 \pm 0.84}$ | $\mathbf{64.36 \pm 0.59}$ |

Table 7: Results for object classification on the AWA2 and aPY dataset. Resutlts for methods with * (asterisk) obtained from Verma et al. (2020), E-PGN from (Yu et al., 2020), APNet from Liu et al. (2020), AREN from (Xie et al., 2019), LGGAA from (Liu et al., 2019b), SGMAL from (Zhu et al., 2019), TGG from (Zhang et al., 2019a), and APN from (Xu et al., 2020). We report the average class-balanced accuracy of the for GCNZ, SGCN, DGP, and ZSL-KG on 5 random seeds and the standard error.

where $\boldsymbol{m}_j \in \mathbb{R}^d$ is the pretrained word embeddings from GloVe.

We learn the context of the mention using two biLSTM with attention modules. The left context $l$ is represented by $\{l_1, l_2, ..., l_s\}$ and the right context $r$ by $\{r_1, r_2, ..., r_s\}$ where $l_i \in \mathbb{R}^d$ and $r_j \in \mathbb{R}^d$ are the word embeddings for the left and the right context. $s$ is the window size for the context. We pass $l$ and $r$ through their separate biLSTM layers to get the hidden states $\overleftarrow{\boldsymbol{h}_l}, \overrightarrow{\boldsymbol{h}_l}$ for the left context and $\overleftarrow{\boldsymbol{h}_r}, \overrightarrow{\boldsymbol{h}_r}$ for the right context.

The hidden states are passed through the attention layer to compute the attention scores. The attention layer is a 2 layer feedforward neural network and computes the normalized attention for each of the hidden states $v_c \in \mathbb{R}^h$:

$$\alpha_i^l = \boldsymbol{W}_\alpha(\tanh(\boldsymbol{W}_e \begin{bmatrix} \overleftarrow{\boldsymbol{h}_i^l} \\ \overrightarrow{\boldsymbol{h}_i^l} \end{bmatrix})) \tag{12}$$

$$a_i^l = \frac{\exp\left(\alpha_i^l\right)}{\sum_i \exp\left(\alpha_i^l\right) + \sum_j \exp\left(\alpha_j^r\right)} \tag{13}$$

The scalar values are multiplied with their respective hidden states to get the final context vector representation $v_c$:

$$\boldsymbol{v}_c = \sum_{i=0}^s a_i^l \boldsymbol{h}_i^l + \sum_{j=0}^s a_j^r \boldsymbol{h}_j^r \tag{14}$$

Finally, we concatenate the context vector $\boldsymbol{v}_c$ and $\boldsymbol{v}_m$ to get the example representation $\boldsymbol{x}$.

## H  FINE-GRAINE ENTITY TYPING EVALUATION

Fine-grained entity typing is a multi-label classification task. We follow the standard evaluation metric introduced in Ling & Weld (2012): Strict Accuracy, Loose Micro F1 and Loose Macro F1.

We denote the set of ground truth types as T and the set of predicted types as P. We use the F1 computed from the precision $p$ and recall $r$ for the evaluation metrics mentioned below:

**Strict Accuracy.** The prediction is considered correct if and only if $t_e = \hat{t_e}$:

$$p = \frac{\sum_{e \in P \cap T} \mathbf{1}(t_e = \hat{t_e})}{|P|} \tag{15}$$

$$r = \frac{\sum_{e \in P \cap T} \mathbf{1}(t_e = \hat{t_e})}{|T|} \tag{16}$$

**Loose Micro.** The precision and recall scores are computed as:

$$p = \frac{\sum_{e \in P} |t_e \cap \hat{t_e}|}{\sum_{e \in P} |\hat{t_e}|} \tag{17}$$

$$p = \frac{\sum_{e \in T} |t_e \cap \hat{t_e}|}{\sum_{e \in T} |\hat{t_e}|} \tag{18}$$

**Loose Macro.** The precision and recall scores are computed as:

$$p = \frac{1}{|P|} \sum_{e \in P} \frac{|t_e \cap \hat{t_e}|}{|\hat{t_e}|} \tag{19}$$

$$r = \frac{1}{|T|} \sum_{e \in T} \frac{|t_e \cap \hat{t_e}|}{|\hat{t_e}|} \tag{20}$$

## I  GRAPH NEURAL NETWORKS ARCHITECTURE DETAILS

| Method | Aggregate | Combine |
|---|---|---|
| ZSL-KG-GCN | $\boldsymbol{a}_v^{(l)} = \mathrm{Mean}\left(\left\{\boldsymbol{h}_u^{(l-1)}, u \in \mathcal{N}(v)\right\}\right)$ | $\boldsymbol{h}_v^{(l)} = \sigma\left(\boldsymbol{W}^{(l)} \boldsymbol{a}_v^{(l)}\right)$ |
| ZSL-KG-GAT | $\alpha_u^{(l)} = \mathrm{Attn}\left(\left\{(\boldsymbol{h}_u'^{(l-1)} || \boldsymbol{h}'^{(l-1)})_v, u \in \mathcal{N}(v)\right\}\right)$ | $\boldsymbol{h}_v^{(l)} = \sigma(\sum_{u=1}^{\mathcal{N}(v)+1} \alpha_u^{(l)} \boldsymbol{h}_u'^{(l-1)})$ |
| ZSL-KG-RGCN | $\boldsymbol{a}_v^{(l)} = \sum_{r \in R} \sum_{j \in N(v)^r} \frac{1}{c_{i,r}} \sum_{b \in B} \alpha_{b,r}^{(l)} \boldsymbol{V}_b^{(l)} \boldsymbol{h}_j^{(l-1)}$ | $\boldsymbol{h}_v = \sigma(\boldsymbol{a}_v + \boldsymbol{W}_s^{(l)} \boldsymbol{h}_v^{(l-1)})$ |
| ZSL-KG-LSTM | $\boldsymbol{a}_v^{(l)} = \mathrm{LSTM}^{(l)}\left(\boldsymbol{h}_u^{(l-1)} \forall u \in \mathcal{N}(v)\right)$ | $\boldsymbol{h}_v^{(l)} = \sigma\left(\boldsymbol{W} \cdot [\boldsymbol{h}_v^{(l-1)} || \boldsymbol{a}_v^{(l)}]\right)$ |

Table 8: Graph Aggregators

ZSL-KG-GCN uses a mean aggregator to learn the neighbourhood structure. ZSL-KG-GAT projects the neighbourhood nodes to a new features $\boldsymbol{h}_u'^{(l-1)} = \boldsymbol{W} \boldsymbol{h}_u^{(l-1)}$. The neighbourhood node features are concatenated with self feature and passed through a self-attention module for get the attention coefficients. The attention coefficients are multiplied with the neighbourhood features to the get the node embedding for the $l$-th layer in the combine function. ZSL-KG-RGCN uses a relational aggregator to learn the structure of the neighbourhood. To avoid overparameterization from the relational weights, we perform basis decomposition of the weight vector into $B$ bases. We learn $|B|$ relational coefficients and $|B|$ weight vectors in the aggregate function and add with the self feature in combine function. ZSL-KG-LSTM uses LSTM as an aggregator to combine the neighbourhood features. The nodes in the graph are passed through an LSTM and the last hidden state is taken as the aggregated vector. The aggregated vector is concatenated with the node's previous layer feature and passed to the combine function to get the node representation.

## J  HYPERPARAMETERS

In this section, we detail the hyperparameters used in our experiments.

### J.1  TRAINING DETAILS

Our framework is built using PyTorch and AllenNLP (Gardner et al., 2017). In all our experiments, we use Adam (Kingma & Ba, 2015) to train our parameters with a learning rate of 0.001. For intent classification, we experiment with a weight decay of 1e-05 and 5e-05. We found that weight decay of 5e-05 gives the best performance overall in intent classification for all the baseline graph aggregators. In intent classification, ZSL-KG use weight decay of 1e-05. We add a weight decay of 5e-05 for the OntoNotes experiments. Finally, all experiments in zero-shot object classification have a weight decay of 5e-04.

| Task | Inp. dim. | Hidden dim. | Attn. dim. | Low-rank dim. |
|---|---|---|---|---|
| Intent classification | 300 | 32 | 20 | 16 |
| Fine-grained entity typing | 300 | 100 | 100 | 20 |

Table 9: Hyperparameters for the biLSTM with attention example encoder in the language related tasks

For intent classification and fine-grained entity typing, we assume a low-rank for the compatibility matrix $W$. The matrix $W \in \mathbb{R}^{d \times d}$ is factorized into $A \in \mathbb{R}^{h \times d}$ and $B \in \mathbb{R}^{d \times h}$ where $h$ is the low-rank dimension. Table 9 summarizes the hyperparameters used in the example encoders which is a biLSTM with attention or a task-specific variant of it.

In fine-grained entity typing, we have two baselines that do not use graph neural networks: OTyper and DZET. OTyper averages the GloVe embedding of 300-dim for the class representations. DZET uses a biLSTM with attention for the class encoder with the same hyperparameters as fine-grained entity typing from Table 9.

### J.2  GRAPH AGGREGATOR SUMMARY

| Task | layer-1 | layer-2 |
|---|---|---|
| Object classification | 2048 | 2049 |
| Intent classification | 64 | 64 |
| Fine-grained entity typing | 128 | 128 |

Table 10: Output dimensions of the graph neural networks in our experiments.

| | layer 1 | | layer 2 | |
|---|---|---|---|---|
| | $d_{(f)}$ | $d_{(p)}$ | $d_{(f)}$ | $d_{(p)}$ |
| AWA2/aPY | 100 | 150 | 1024 | 1024 |
| SNIPS-NLU | 150 | 150 | 32 | 32 |
| OntoNotes | 150 | 150 | 32 | 64 |
| BBN | 250 | 150 | 32 | 64 |

Table 11: The hyperparameters used in our Transformer Graph Convolutional Network.

Table 10 describes the output dimensions of the node embeddings after each graph neural network layer. ZSL-KG-GCN, DGP, GCNZ, and SGN are linear aggregators and learn only one weight matrix in each of the layers. ZSL-KG-GAT learns a weight matrix for the attention where $W_a \in \mathbb{R}^{2d_{(1)} \times 1}$ and uses LeakyReLU activation in the attention. LeakyReLU has a negative slope of 0.2. ZSL-KG-RGCN learns $B$ bases weight vectors in the baseline. We found that $B = 1$ performs the best for fine-grained entity typing and object classification. For intent classification, we use 10 bases, i.e.,

$B = 10$. In intent classification and fine-grained entity typing, the non-linear activation function after the graph neural network layer is ReLU and in object classification the activation function is LeakyReLU with a negative slope of 0.2.

ZSL-KG with the Transformer Graph Convolutional Network is a complex architecture with numerous parameters. In our transformer module, there are five hyperparameters - input dimension $d_{(l-1)}$, output dimension $d_{(l-1)}$, feedforward layer hidden dimension $d_{(f)}$, and projection dimension $d_{(p)}$. The input dimension and output dimensions are the same in the aggregator. We tuned the hyperparameters on the held out validation classes for object classification and intent classification. For fine-grained entity typing we manually tuned the hyperparameters. Table 11 details the hyperparameters used in our experiments for all the datasets.

