# OpenReview forum: "Zero-Shot Learning with Common Sense Knowledge Graphs"
_ICLR.cc/2021/Conference — Reject_

### Official Review · AnonReviewer1 · 2020-10-28
**Official Blind Review #1**

**Rating:** 7
**Confidence:** 5

**Review:**

Summary

This paper tackles zero-shot learning by leveraging the large-scale knowledge graph i.e., ConceptNet to propagate the knowledge learned from seen classes to unseen classes. The authors propose a novel propagation rule that aggregates node embeddings by the self-attention technique. It is infeasible to run GCN on such large-scale knowledge graph. Therefore they reduce the knolwedge graph by adopting a neighborhood sampling strategy based on random walks. The method is evaluated on multiple zero-shot learning tasks including object classification, intent classification and fine-grained entity typing. The SOTA results are achieved.

Pros

-This paper is well motivated. The idea of leveraging a large-scale common sense knowledge graph that contains richer semantic relationship is reasonable and novel. I agree that previous GCN methods for zero-shot learning are limited by the wordnet.

-The self-attention or transformer-based aggregator is novel as well. Applying such non-linear aggregation makes graph capture complex relationships between nodes. Therefore, I believe the technical contribution is significant.

-The evaluation is conducted on multiple zero-shot learning tasks and the results show that the proposed approach generalizes  well to different tasks.

-The paper is written well and easy to follow in general, although some important details and justification of the results are missing (I will points out the issues below).

Cons

-I am particularly not convinced by the object classification results (Table 1). This paper only reports results on two small datasets i.e., AWA2 and aPY, in the zero-shot learning setting, while the GCN baselines i.e., GCNZ, SGCN and DGP, are all evaluated on the large-scale ImageNet in both zero-shot and generalized zero-shot learning settings. I strongly suggest the authors to compare with GCNZ, SGCN and DGP on the ImageNet.

-The results on the BBN dataset (Table 3) are not impressive. The proposed approach performs much worse than DZET
in two metrics i.e., Loose Mic. and Loose Mac. The authors fail to give some justification for these bad results.

-The neighborhood sampling technique is rarely evaluated. I believe it is interesting to ablate different sampling strategies e.g., uniformly ramdom sampling, importance sampling (Chen et al., 2018a) and random walks. Implementation details for sampling are not provided either which makes it hard to reproduce the results.

-Writing can be further improved in the following perspectives. A discussion on the difference between the common sense knowledge graph and the standard knowledge like WordNet adopted by previous ZSL works should be included. The justification for the bad results on the BBN dataset is missing. There are not enough implementation details in order to reproduce the results e.g., what are the text features used for the intent classification? How many nodes are selected in the graph sampling step?

Minor questions

-In the last paragraph of Section 4.1, I do not get why other GCN approaches are referred as general-purpose methods?

Justification of the rating

-Overall, I think this paper has significant technical contributions. However, due to the issues I have pointed out above, my initial score is only 6 (marginally above the acceptance threshold.). If the authors can address my concerns, in particular, providing better results on the ImageNet dataset, I will be very happy to upgrade my score.

----------------------------------------------------
Post-rebuttal

Thanks for the new ZSL and GZSL results on ImageNet. The results are convincing to me. My other concerns are properly addressed too. I read the concerns from R4 and R5. To my knowledge, using common sense knowledge graph and the GCN with self-attention are novel in the zero-shot learning literature. I decide to increase my score to an accept.

---

> ### Author Response · Authors · 2020-11-17
> **Thank you for your feedback!**
>
> We really appreciate your feedback on our paper. As suggested, we have added updated experiments on the ImageNet dataset. We report the state-of-the-art on both the traditional zero-shot learning and the generalized zero-shot learning settings (Table 3). ZSL-KG offers relative improvements as high as 9.1% compared to the best reported results on the dataset. We have also included a discussion section describing the differences between common sense knowledge graphs and ImageNet for ZSL. Finally, we have included several suggestions that you have pointed out in your review.
>
> **\#\#** *“This paper only reports results on two small datasets i.e., AWA2 and aPY, in the zero-shot learning setting”*
>
> We would like to point out that our results on AWA2 and aPYdo not require any training images from the two datasets to achieve state-of-the-art performance. In all cases, the model is trained on ImageNet 1K. Additionally, we have updated the results for ImageNet and ZSL-KG achieves state-of-the-art results on both traditional zero-shot learning and generalized zero-shot learning settings.
>
>
> **\#\#** *“The proposed approach performs much worse than DZET in two metrics i.e., Loose Mic. and Loose Mac. The authors fail to give some justification for these bad results.”*
>
> You are correct in pointing out that our method performs relatively poorly on Loose Micro and Loose Macro. Both these metrics allow the model to predict false positives for labels. DZET overpredicts compared to the ZSL-KG but with a lower precision that reduces its overall strict accuracy. We include this point in the paper.
>
>
> **\#\#** *“I believe it is interesting to ablate different sampling strategies e.g., uniformly random sampling, importance sampling (Chen et al., 2018a) and random walks.”*
>
> Thank you for pointing out that we could ablate over different sampling techniques. ConceptNet is an extremely noisy graph and we observed that random-walk based sampling performed better random sampling of the neighbors. We agree this is an important area for future work.
>
> **\#\#** *“Implementation details for sampling are not provided either which makes it hard to reproduce the results.”*
>
> Due to the lack of space in the main paper, we included random-walk details in the appendix (Section I.2).
>
>
> **\#\#** *“A discussion on the difference between the common sense knowledge graph and the standard knowledge like WordNet adopted by previous ZSL works should be included.”*
>
> Thank you for your suggestion. We have included the differences between common sense knowledge graphs and wordnet for ZSL in Section 3.1 of the paper.
>
> **\#\#** *“There are not enough implementation details in order to reproduce the results ”*
>
> We have included the architecture details, hyperparameters, and other details to reproduce the paper in the appendix. Please let us know if there are other details you would like to see included. In addition, the code is available in the supplementary material and will be released publicly.
>
> **\#\#** *“why other GCN approaches are referred as general-purpose methods”*
>
> We call them general-purpose because it is possible to apply their work for other tasks, even if they haven’t been used that way before. We wanted a thorough comparison of other related works across multiple tasks. We reproduce their method with the ImageNet graph on all three types of tasks and show that our method ZSL-KG outperforms the best of these methods by 5.3 accuracy points on average.

---

> > ### Comment · AnonReviewer1 · 2020-11-17
> > **Random-walk details**
> >
> > Thanks for the response and new results. My concerns are addressed.
> >
> > But I cannot find the random-walk details in the appendix (Section I.2) since there is no Section I.2. Could you point me to the correct section? It would be good if the codes including sampling of the graph can be released.

---

> > > ### Author Response · Authors · 2020-11-20
> > > **Random-walk details**
> > >
> > > Sorry for the mixup. The section number was according to the older organization of the appendix. You can find some details about random-walk sampling in Section 4 under Setup. We include further details in Section C2 of the appendix.
> > >
> > > The code for sampling the graph for TrGCN can be found in our supplementary material in `object_classification/gnn/transformer_agg.py`  on line 68. Please let us know if you would like us to add anything else.

---

### Official Review · AnonReviewer4 · 2020-10-29
**Novelty of the proposed model is limited and more analyses are needed in experiments**

**Rating:** 4
**Confidence:** 4

**Review:**

The paper proposes to tackle the zero-shot learning problem by learning class representation from commonsense knowledge graphs. To capture knowledge in graphs, the paper proposes to perform Transformer-based aggregation.

Pros:
- The motivation of using commonsense knowledge graphs to enhance zero-short learning is interesting (although it has been explored in the previous work).
- The paper is in general clearly written and easy to follow.

Cons:
- In term of methodology, the novelty of paper is very limited---it uses Transformer to aggregate knowledge over graphs. This is pretty straightforward, and the previous work (e.g., (Kampffmeyer et al.)) has tried Graph Attention Networks (GAT)-based models for zero-shot learning. Transformer and GAT are very similar, particularly in the setup of this work (e.g., without sequence/position embedding and the graphs are not fully connected). The differences claimed in this paper between GAT-based and Transformer-based models are rather marginal.

- The empirical comparison to GAT-based models in the experiments is not clear enough. For example, Table 4 shows that the proposed model is better than ZSL-KG-GAT, but detailed analyses were not provided to make this clearer to help understand why that is the case (and hence better understand the contributions of this paper). For example, whether the performance differences are related to the use of layer normalization, the two-layer feed-forward nets in equation (4), or other reasons? Previous work suggested DGP is better than GAT-based models (Kampffmeyer et al.) and in this current submission GAT and transformer-based models are better. More discussions will be helpful.

- It will be more helpful if the paper describes more details about the models in comparison, e.g., details about ZSL-KG-GAT such as the setup of multi-heads.

More comments:
- The title of the paper may be made more specific, particularly given much work has been done by using commonsense/knowledge graphs for zero-short learning.

---

> ### Author Response · Authors · 2020-11-17
> **Transformers vs. Graph Attention Networks**
>
> Thank you for your comments!
>
> **\#\#** *“In term of methodology, the novelty of paper is very limited---it uses Transformer to aggregate knowledge over graphs.”*
>
> We respectfully disagree on this point. There are two main methodology contributions. First, to the best of our knowledge, we are the first to learn class representations from nodes in common sense knowledge graphs without additional human input. Please also see the message addressed to all reviewers and the response to AnonReviewer5 for additional discussion of this point. Second, the use of Transformers as a GNN aggregator is significant because, to our knowledge, it is the first proposed aggregator that is both non-linear and permutation invariant. Please also see the review of AnonReviewer1 who agrees with these claims.
>
> **\#\#** *“the previous work (e.g., (Kampffmeyer et al.)) has tried Graph Attention Networks (GAT)-based models for zero-shot learning”*
>
> We would like to clarify that graph attention networks (GAT) compared in Kampffmeyer et al. is different in several ways. First, ZSL-KG uses transformer graph convolutional network (TrGCN), a novel architecture introduced in this work that creates non-linear combinations of the nodes’ neighborhoods. GAT and TrGCN are two different graph neural network architectures. Second, GAT based models compared in Kampffmeyer et al. learn class representations from the ImageNet graph. Finally, ZSL-KG-GAT in Table 5 is an ablated version of ZSL-KG, so it still uses our conceptual contribution of subsampling a common sense knowledge graph to create class representations.
>
> **\#\#** *“The differences claimed in this paper between GAT-based and Transformer-based models are rather marginal.”*
>
> As described above, graph attention networks (GAT) and transformer graph convolutional networks (TrGCN) are different in the way they aggregate information over the graph structure. GAT computes the aggregated vector by computing the linear combination of the node features in the neighbourhood. TrGCN learns a transformer-based aggregator and computes a non-linear combination of the node features in the neighbourhood, followed by a pooling function to get the aggregated vector which leads to the difference in architecture.
>
>
> **\#\#** *“The empirical comparison to GAT-based models in the experiments is not clear enough… whether the performance differences are related to the use of layer normalization, the two-layer feed-forward nets in equation (4), or other reasons?”*
>
> As noted in the paper, the feedforward neural network layer in TrGCN adds non-linearity to the node neighbourhood features, and the self-attention layer computes the weighted combination of the non-linear features by querying for each node in the neighborhood, which further increases the expressivity of TrGCN.
>
>
> **\#\#** *“Previous work suggested DGP is better than GAT-based models (Kampffmeyer et al.) and in this current submission GAT and transformer-based models are better.”*
>
> We would like to highlight reasons why ZSL-KG performs better than DGP. First, ZSL-KG learns class representations with common sense knowledge graphs, whereas DGP learns class representations using the ImageNet graph. Common sense knowledge graphs offer rich associations that contribute to the improvement in performance. Second, previous work has not compared DGP on a wide range of tasks in both language and vision. Our results show that ZSL-KG outperforms DGP on all six benchmark zero-shot datasets. Finally, GAT referred to as ZSL-KG-GAT is a variant of ZSL-KG and uses our conceptual contribution of learning from common sense knowledge graphs, which helps explain why it also outperforms DGP.
>
>
> **\#\#** *“The title of the paper may be made more specific, particularly given much work has been done by using commonsense/knowledge graphs for zero-short learning.”*
>
> Our work is the first paper that learns class representations only from common sense knowledge graphs for zero-shot learning tasks without any additional human input. We show that it works well on both language and vision tasks. It would be great if you could point us to more related work in the area.

---

### Official Review · AnonReviewer5 · 2020-11-04
**Limited technical contribution and experiment results not convincing**

**Rating:** 4
**Confidence:** 5

**Review:**

The submission proposed to leverage a commonsense knowledge graph and an attention GNN based model to aggregate the node features on the graph for the problem of zero-shot learning.

The main concern is the technical contribution is limited:

- The authors mention "Existing methods for zero-shot object classification such as GCNZ and DGP cannot be
adapted to common sense knowledge graphs as they do not scale to large graphs or require a directed
acyclic graph such as WordNet." Can the authors elaborate on why GCNZ and DGP can not scale to large knowledge graphs? To my understanding, GCNZ and DGP use WordNet knowledge graphs which are large-scale. And [a] also proposed to leverage GNN structure and knowledge graph, can the authors explain more comparisons with respect to this paper?

- The proposed GAT model is similar to the previous works, i.e., GCNZ, and DGP. To me, the model GNN and attention (Transformer in the paper) model have already been discussed in these two works and the only difference seems to be the knowledge graph, where the previous paper uses the WordNet while the current submission uses the ConceptNet. (I think WordNet is a more straightforward knowledge graph format, can the authors explain why extra efforts on making a ConceptNet is better?) Also, [e] this paper has already proposed to leverage ConcepNet and GNN structures for ZSL, can the authors explain the difference with it please?

- The authors claim the structure of Transformer while actually, it is the structure of dot product attention. The appliance of the attention of ZSL has been a lot, such as [a,b,c,d]. So I think the technical contribution is limited and it is better to also include the discussions with these works.

Based on the previous three points, I believe the technical contribution is limited for this submission.

The experiments do not support the claims:

- The authors claim the model can scale to large size while in the experiment section,  the results on ImageNet is not reported. The most related two works GCNZ and DPN both reported their results on ImageNet, thus i believe the performance result on this dataset is necessary.

- I am expected to see performance on the generalized zero-shot learning since this setting is more practical and includes the prediction on both seen and unseen classes. The most recent papers including the previous works listed in the experiment table have the results on GZSL, so do the GCNZ and DPN. Thus, I think the results on GZSL are also necessary to support the claims in this submission.


Writing:
Even the basic format is not well organized, such as the tables are out of columns and spelling errors.

Related works:
The current related works section is a little bit precise, and I would recommend the authors to include the related works mentioned in my previous reviews and discuss the difference and how this submission is different from previous works. Since there is plenty of previous literature that is somewhat related to the current submission, a more detailed explanation of the relationships to previous works is recommended.

[a] Attribute Propagation Network for Graph Zero-shot Learning, AAAI 2021

[b] Attentive Region Embedding Network for ZSL, CVPR19.

[c] Semantic-Guided Multi-Attention Localization for ZSL, NIPS19.

[d] Attribute Attention for Semantic Disambiguation in ZSL, ICCV19.

[e] TGG: Transferable Graph Generation for Zero-shot and Few-shot Learning

---

> ### Author Response · Authors · 2020-11-17
> **Clarifications on ZSL-KG (1/2)**
>
> Thank you for your thoughtful comments!  We would like to address your questions in the review.
>
> **\#\#** *“Can the authors elaborate on why GCNZ and DGP can not scale to large knowledge graphs? To my understanding, GCNZ and DGP use WordNet knowledge graphs which are large-scale.”*
>
> GCNZ requires the entire graph structure during training, and the method cannot accommodate changing the graph structure at test time, which is necessary for inductive zero-shot learning. ConceptNet has over 8 million nodes and 21 million edges. On the other hand, the ImageNet subset of the WordNet graph used in GCNZ has about 32,000 nodes and about 65,000 edges. It would be impractical and infeasible to train a graph neural network with GCNZ on a common sense knowledge graph with millions of edges. Furthermore, DGP requires a directed acyclic graph or parent-child relationship in the graph. We are not restricted to parent-child relationships in a common sense knowledge graph.
>
> **\#\#** *“And [a] also proposed to leverage GNN structure and knowledge graph, can the authors explain more comparisons with respect to this paper?”*
>
> You are correct that [a] uses a knowledge graph and graph neural network for zero-shot learning, but there are several key differences. ZSL-KG learns to map nodes in a knowledge graph to class representations without any other human input. In contrast, the method in [a] learns to transfer hand-engineered attributes to new nodes in the graph. Finally, our results on AWA2 and aPY outperform [a] by 10.08 and 23.06 absolute accuracy points. We include comparison of ZSL-KG on AWA2 and aPY with other related work in Table 7 (Appendix E).
>
> Please also note that [a], [b], [c],[d], and [e] all use hand-engineered attributes. As discussed in our paper, this is a limiting assumption. Our focus is on methods that can map nodes in graphs to class representations without any additional human input.
>
>
> **\#\#** *“The proposed GAT model is similar to the previous works, i.e., GCNZ, and DGP.”*
>
> We would like to clarify this point. First, ZSL-KG learns the graph structure with our proposed transformer graph convolutional networks (TrGCN) which is not the same as the graph attention networks (GAT). The TrGCN model learns a non-linear combination of its node neighbours with a transformer-based aggregator whereas GAT learns a linear combination. Second, GCNZ and DGP make restrictive assumptions that render them applicable only to WordNet or ImageNet, as described above.
>
> **\#\#** *“I think WordNet is a more straightforward knowledge graph format, can the authors explain why extra efforts on making a ConceptNet is better?”*
>
> Our paper goes beyond object classification and achieves state-of-the-art performance on language tasks as well. Learning node representations with ConceptNet helps in learning a richer representation for the classes. Our results on intent classification and fine-grained entity typing show an average improvement of 6.51 and 4.56 accuracy points over the best performing methods using the ImageNet graph.
>
> **\#\#** *“Also, [e] this paper has already proposed to leverage ConceptNet and GNN structures for ZSL, can the authors explain the difference with it please?”*
>
> Thank you for suggesting another related work [e] that uses ConceptNet in their zero-shot learning method. It is critical to note that they use ConceptNet to establish edges between the class attributes in the training rather than learn class representations from the graph for zero-shot learning.  Furthermore, our results on AWA2 and aPY outperform their method on zero-shot learning on AWA2 and aPY by about 1 accuracy point on both datasets (Table 2 from their paper). We have added a citation to the manuscript.
>
> Also note that requiring explicit attributes restricts the applicability of the method. (Please also see above.)
>
> **\#\#** *“The appliance of the attention of ZSL has been a lot, such as [a,b,c,d].”*
>
> [a], [b],[c], and [d] methods use attention over the image regions. On the other hand, our method uses attention over the graph. Furthermore, the focus of our work is not the example encoder or the image encoder. Our work focuses on the class encoder (TrGCN), which can potentially complement the other methods like [a], [b], [c], and [d].
>
> **\#\#** *“The authors claim the model can scale to large size while in the experiment section, the results on ImageNet are not reported.”*
>
> We appreciate you for pointing out that our method needs to be evaluated on the ImageNet dataset as well. However, we would like to clarify that “large scale” is referring to in the context of the graphs used to represent concepts, not the size or number of classes in the test set. Nonetheless, we report the state-of-the-art on the ImageNet dataset and achieve relative improvements as high as 9.1% on the dataset (Table 3).

---

> > ### Author Response · Authors · 2020-11-17
> > **Clarifications on ZSL-KG (2/2)**
> >
> > **\#\#** *“The most recent papers including the previous works listed in the experiment table have the results on GZSL, so do the GCNZ and DPN.”*
> >
> > Thank you for suggesting to report results on generalized zero-shot learning (GZSL). 2 out of the 5 datasets in our original paper (the fine-grained entity typing tasks) are GZSL problems. We have included updated experiment results on the ImageNet dataset. ZSL-KG achieves state-of-the-art on GZSL evaluation and offers relative improvements as high as 9.1% compared to best results on the dataset.
> >
> > **\#\#** *“tables are out of columns and spelling errors.”*
> >
> > We have updated the table formatting in the manuscript. We will proofread the manuscript again, and we would appreciate pointing out examples of spelling errors.
> >
> > **\#\#** *“I would recommend the authors to include the related works mentioned in my previous reviews and discuss the difference and how this submission is different from previous works.”*
> >
> > Thank you for your suggestion on our related works section. We have updated related works in our manuscript.

---

### Author Response · Authors · 2020-11-17
**To all reviewers**

Thank you to all the reviewers for your thoughtful and helpful comments and questions. In addition to more detailed individual replies, we summarize the main clarifications and largest changes to the manuscript here. Please see the individual responses for more detailed discussions and changes.

**Relationship with other work on ZSL**
AnonReviewer4 and AnonReviewer5 asked about the relationship with other works on zero-shot learning. To the best of our knowledge, our work is the only one that learns to map nodes in a common sense knowledge graph on the scale of ConceptNet to class representations, without any additional human input. All other work of which we are aware, including references suggested by the reviewers, (1) require hand-engineered attributes for seen and unseen classes in addition to the knowledge graph, and/or (2) are limited to the (orders of magnitude smaller) ImageNet subset of the WordNet graph, which is not a common sense knowledge graph.

We also note that AnonReviewer1 says “This paper is well motivated. The idea of leveraging a large-scale common sense knowledge graph that contains richer semantic relationships is reasonable and novel. I agree that previous GCN methods for zero-shot learning are limited by the wordnet.”

**Relationship with other work on GNNs**
AnonReviewer4 and AnonReviewer5 asked about the relationship with other works on graph neural networks. ZSL-KG includes a novel transformer graph convolutional network (TrGCN). TrGCN learns to create non-linear combinations of neighborhood nodes by applying self-attention to the entire neighborhood. This aggregator also queries for each node in the neighborhood. This is not the same as using GAT, which creates a linear combination, where the weight of each neighbor is determined by attention. TrGCN is a distinct contribution because it is the only GNN aggregator proposed in the literature to our knowledge that is both non-linear and permutation invariant.

We also note that AnonReviewer1 says “The self-attention or transformer-based aggregator is novel as well. Applying such non-linear aggregation makes graph capture complex relationships between nodes. Therefore, I believe the technical contribution is significant.”

**Additional experiments**
AnonReviewer1 and AnonReviewer5 asked to see more evaluation on unseen ImageNet classes. We have added these results. ZSL-KG on traditional zero-shot learning reports the state-of-the-art and achieves relative improvements as high as 7.9% compared to the best-reported results. AnonReviewer5 asked to see evaluations on the generalized ZSL setting. We note that 2 out of the 5 datasets (fine-grained entity typing) in the original manuscript are generalized ZSL tasks, and we have also added generalized ZSL results for ImageNet. Therefore, half of the experiments now have generalized ZSL results. In the generalized zero-shot learning setting, we report relative improvements as high as 9.1% compared to the best results on the dataset.

---

> ### Comment · AnonReviewer1 · 2020-11-17
> **Concerns are addressed, recommand to accept**
>
> Thanks for the new ZSL and GZSL results on ImageNet. The results are convincing to me. My other concerns are properly addressed too. I read the concerns from R4 and R5.  To my knowledge, using common sense knowledge graph and the GCN with self-attention are novel in the zero-shot learning literature.  I am happy to discuss with R4 and R5 if I miss something.
>
> I will increase my score to an accept.
>
> Minor point: the authors mentioned that on ImageNet, "ZSL-KG ... achieves relative improvements as high as 7.9% compared to the best-reported results". I am wondering where this 7.9% comes from. I did not see such significant improvement in Table 3.

---

> ### Author Response · Authors · 2020-11-20
> **ImageNet updates**
>
> To clarify the statement about ImageNet scores, 7.9% was the highest relative improvement versus the next best performing method, which corresponds to an absolute improvement was 1.53 points (hit@5).
>
> We've also added evaluation on all ImageNet classes in Table 3. We report a new state-of-the-art on this task as well. Compared to the previous state-of-the-art methods, ZSL-KG has an absolute improvement of up to 2.3 points (6.6% relative improvement) and a relative improvement of up to 20% (0.3 absolute improvement).

---

### Decision · Program_Chairs · 2021-01-07
**Final Decision**

**Decision:**

Reject

**Comment:**

The submission proposes to leverage a commonsense knowledge graph and an attention GNN based model to aggregate the node features on the graph for the problem of zero-shot learning. It received three reviews two of them recommending rejection, and another review was initially borderline however they moved to acceptance after the rebuttal. The meta reviewer finds that the paper is not yet ready for publication and recommends rejection based on the following observations.

Although the model is interesting, as agreed by the reviewers the initial version of the paper fell short on convincingly evaluating the method, e.g. generalized zero-shot learning (GZSL) setting as pointed out by R5, ImageNet and small scale dataset results as pointed out by R1. Similarly, the main paper (without the annexes) has been found to fall short on providing enough details of the model as pointed out by R1.

The authors ran additional experiments during the rebuttal phase which showed some promise, however one more review round may be necessary to carefully validate these results. As R1 pointed out, the paper only reports results on two small datasets i.e., AWA2 and aPY, which contain classes similar to ImageNet. It would be interesting to observe the behaviour of the model on more challenging scenarios on other publicly available benchmark datasets of fine-grained nature whose distribution are far from ImageNet. This would indicate the generalisation ability of the model.

Furthermore, as pointed out by R1, moving the details on the implementation and the architecture details from appendix and from python scripts to the main paper may be beneficial. However, this would end up significantly extending the paper. Hence, one more review round may be necessary for this paper.